# Birhythmic Analog Circuit Maze: A Nonlinear Neurostimulation Testbed

**DOI:** 10.3390/e22050537

**Published:** 2020-05-11

**Authors:** Ian D. Jordan, Il Memming Park

**Affiliations:** 1Department of Applied Mathematics and Statistics, Stony Brook University, Stony Brook, NY 11794, USA; ian.jordan@stonybrook.edu; 2Institute for Advanced Computing Science, Stony Brook University, Stony Brook, NY 11794, USA; 3Department of Neurobiology and Behavior, Stony Brook University, Stony Brook, NY 11794, USA

**Keywords:** dynamical systems, bistability, birhythmic, analog circuit, neurostimulation

## Abstract

Brain dynamics can exhibit narrow-band nonlinear oscillations and multistability. For a subset of disorders of consciousness and motor control, we hypothesized that some symptoms originate from the inability to spontaneously transition from one attractor to another. Using external perturbations, such as electrical pulses delivered by deep brain stimulation devices, it may be possible to induce such transition out of the pathological attractors. However, the induction of transition may be non-trivial, rendering the current open-loop stimulation strategies insufficient. In order to develop next-generation neural stimulators that can intelligently learn to induce attractor transitions, we require a platform to test the efficacy of such systems. To this end, we designed an analog circuit as a model for the multistable brain dynamics. The circuit spontaneously oscillates stably on two periods as an instantiation of a 3-dimensional continuous-time gated recurrent neural network. To discourage simple perturbation strategies, such as constant or random stimulation patterns from easily inducing transition between the stable limit cycles, we designed a state-dependent nonlinear circuit interface for external perturbation. We demonstrate the existence of nontrivial solutions to the transition problem in our circuit implementation.

## 1. Introduction

Multistability is a widespread phenomenon in the field of dynamical systems where a system exhibits multiple stable states or more generally attractors [1]. Appearing in nearly all disciplines of natural science and engineering, including biology [2,3], chemistry [4,5], electronics [6,7], fluid mechanics [8,9], genetics [10,11], and physics [12,13], researchers have shown substantial interest in such behavior [1]. In neuroscience, computations are thought to be implemented by multistable dynamical systems, and recent experimental and methodological advances have generated renewed interests [14,15,16,17,18,19]. The multiple attractors within these dynamical systems seem to underlie a wide array of functions, including sensory perception [20], motor function [21], and cognition [22,23], as well as dysfunctions, such as movement disorders [24], epilepsy [25], and disorders of consciousness [26,27]. We hypothesized that multistability underlie some dynamical neurological diseases such that manifested symptoms are fundamentally due to the inability to naturally transition from one basin of attraction to another. Under this hypothesis, neurostimulation techniques provide a means to perturb neural systems to assist transition between attractors as a treatment option.

Open-loop electrical stimulation therapies have shown remarkable successes, most notably with Parkinson’s disease [28]. However, open-loop strategies are likely to be insufficient for the general induction of attractor transitions that manifest complex nonlinear dynamics and non-trivial stimulus induced perturbations. For example, high-amplitude low-frequency signals, such as those dominant in disorders of consciousness [27], suggest the existence of strong attractor dynamics which may require a sophisticated feedback control system to transition out [29]. This sets the stage for the next-generation closed-loop neural stimulators that can intelligently learn to induce attractor transitions. The added complexity of the closed-loop stimulation systems calls for a platform to develop and test their efficacy.

In this paper, we aimed to develop a hardware platform by which these intelligent stimulation algorithms could be tested and validated on. Clinically, this platform can serve as an initial stepping stone, prior to the use of animal models and human patients, in demonstrating the successful use of a stimulation regime under our hypothesis for a given application. As is a common method for realizing a dynamical system physically, an analog electronic circuit exhibiting the desired dynamics is constructed [30,31,32]. Due to the nature of widely used electrical stimulation for neurological implants, this medium will serve well as a testbed. While such a system can be developed as a software package, neurostimulators have an analog interface, where typically current is injected into the neural tissue. By keeping our platform entirely analog, we maintain this interface. Furthermore, the analog nature of the brain ensures the existence of inherent noise and imperfections in the system dynamics. We would like to implement these imperfections here as a result of the medium by which the platform is designed on, rather than needing to over-complicate our model. This added variability will require a stimulation algorithm to be more robust.

To reduce unnecessary complexity, we constructed our system to demonstrate the simplest form of oscillatory multistability, birhythmicy. More specifically, the system will simultaneously exhibit two self-exciting limit cycles of notably different frequencies. Given that the state of the system is sufficiently close to one of the two attractors, an intelligent stimulation algorithm can be tested by trying to perturb the system into the other basin of attraction. While this may seem like a substantial simplification of global brain dynamics for typical neurological function, when viewing brain activity at different spatiotemporal scales under specific conditions far fewer attracting states may be present. Two examples are patients under deep anesthesia or within a comatose state. Under such conditions, there appears to be coherent synchronization across large brain regions [26,27]. Such homogeneity would vastly simplify the underlying dynamic behavior, resulting in far fewer attractors. Under the right lens, our developed model becomes more immediately applicable.

In the following section, we derive the system from the general continuous-time dynamical system underlying the gated recurrent unit (GRU), a commonly used recurrent neural network architecture [33,34]. In Section 3, we discuss the details of the circuit design and present the results of the physical realization. In Section 4, we discuss the addition and design of a nonlinear circuit, state dependent on the system described in Section 2 and Section 3, by which external stimulation is interfaced. The addition of this nonlinear *stimulator circuit* will ensure random or periodic stimulation patterns will be ineffective in inducing transitions between the two attractor states.

## 2. Birhythmic Dynamics in 3-Dimensions

Our goal was to find a simple bistable dynamical system where each attractor corresponds to a periodic orbit. We draw from the recurrent neural network literature on simple forms of stable limit cycles. Specifically, we utilize the autonomous continuous-time gated recurrent unit (ct-GRU) [33,34] formulation, which can be represented as follows: (1)$$\begin{array}{cccc} \dot{h} & =(1-s(t))⊙(T(t)-h(t))\; & (\mathrm{hidden}\;\mathrm{state}) & , \end{array}$$(2)$$\begin{array}{cccc} s(t) & =\sigma (U_{s}h\left(t\right)+b_{s})\; & (\mathrm{update}\;\mathrm{gate}) & , \end{array}$$(3)$$\begin{array}{cccc} r(t) & =\sigma (U_{r}h\left(t\right)+b_{r})\; & (\mathrm{reset}\;\mathrm{gate}) & , \end{array}$$(4)$$\begin{array}{cccc} T(t) & =\tanh(U_{h}\left(r\left(t\right)⊙h\left(t\right)\right)+b_{h})\; & \; & , \end{array}$$
where $h\left(t\right)\in R^{d}$ is the state of the system, $U_{s},U_{r},U_{h}\in R^{d\times d}$ are the parameter matrices, $b_{s},b_{r},b_{h}\in R^{d}$ are the bias vectors, ⊙ represents the Hadamard product, and $\sigma \left(z\right)=1/\left(1+e^{-z}\right)$ is the element-wise logistic sigmoid function. For a given set of parameters, fixed points of the system exist where $\dot{h}=0$. Since 1−s(t)>0,∀s, this term does not influence the roots of the right-hand side of Equation (Equation 1). Therefore, s(t) can only affect the speed of the flow and, in turn, can be neglected when choosing a set of parameters for the system to enact a desired structure of attractors [34]. Note that, if the parameters of r(t) have been set to zero, the ct-GRU architecture simplifies to the classic ct-tanh-RNN if the parameters of s(t) are also set to zero.

In previous work [34], we have shown that, for d=2, the ct-GRU is capable of expressing a single limit cycle (attracting closed orbit) in phase space under the following set of parameters:(5)$$U_{r},b_{r},b_{h}=0,\;U_{h}=3\left[\begin{array}{cc} \cos\alpha & -\sin\alpha \\ \sin\alpha & \cos\alpha \end{array}\right],$$
where $\alpha \in S_{0}$ and $S_{0}⊇\left(\frac{\pi }{21},\frac{\pi }{3.8}\right)$. The phase portrait depicting this behavior for $\alpha =\frac{\pi }{5}$ can be seen in Figure 1, where $h\equiv {\left[\begin{array}{cc} x & y \end{array}\right]}^{T}$.

Extending the system to 3-dimensions allows for the simultaneous existence of two limit cycles in phase space under a single set of parameters. More specifically, the addition of a third dimension enables us to mirror any attractor structure representable for d=2 across an unstable manifold on the plane defined by the original two dimensions in $R^{3}$. This behavior is depicted in Figure 2A, where now $h\equiv {\left[\begin{array}{ccc} x & y & z \end{array}\right]}^{T}$, and the parameters are set as follows:(6)$$U_{r},b_{r},b_{h}=0,\;U_{h}=3\left[\begin{array}{ccc} \cos\frac{\pi }{5} & -\sin\frac{\pi }{5} & 0\\ \sin\frac{\pi }{5} & \cos\frac{\pi }{5} & 0\\ 0 & 0 & 1 \end{array}\right].$$

As stated before, s(t) only acts to adjust the speed of phase flow. If $U_{s},b_{s}=0$, the periods of both limit cycles are equal. As a means to easily decouple the two frequencies of oscillation, the velocity of flow may be made dependent on its vertical position with respect to the *z*-axis. While the range of the logistic sigmoid function has the benefit of always being defined on (0,1), it may produce inaccurate results when physically realized along its tails. Furthermore, any function that is strictly positive and sufficiently well-behaved on the region of phase space we are interested in will work under this context.

For simplicity, we redefine s(t) linearly as $s\left(t\right)=U_{s}h\left(t\right)+b_{s}$. We then note that Equation () is asymptotically bound to ${\left[-1,1\right]}^{d}$. To account for potential error in the electronic realization, we set s(t) such that its output remains strictly positive on (−1.5,1.5)×(−1.5,1.5)×R. The results of this linear update-gate are demonstrated in Figure 2B,C and achieved under the following set of parameters in conjunction with those of Equation (Equation 6):(7)$$b_{s}=\left[\begin{array}{c} -0.5\\ -0.5\\ 0.5 \end{array}\right],U_{s}=\left[\begin{array}{ccc} 0 & 0 & 1\\ 0 & 0 & 1\\ 0 & 0 & 0 \end{array}\right].$$
To better grasp the dynamical system depicted in Figure 2 to be later realized, we can rewrite Equations (Equation 1)–(4) explicitly in terms of x,y and *z* with our chosen parameters from Equations (Equation 6) and (Equation 7) as follows: (8)$$\begin{array}{cc} \dot{x} & =\left(z-\frac{3}{2}\right)\left[x-\tanh\left(x\cdot \frac{3}{2}\cos\frac{\pi }{5}-y\cdot \frac{3}{2}\sin\frac{\pi }{5}\right)\right], \end{array}$$(9)$$\begin{array}{cc} \dot{y} & =\left(z-\frac{3}{2}\right)\left[y-\tanh\left(x\cdot \frac{3}{2}\sin\frac{\pi }{5}+y\cdot \frac{3}{2}\cos\frac{\pi }{5}\right)\right], \end{array}$$(10)$$\begin{array}{cc} \gamma \dot{z} & =-\frac{1}{2}\left[z-\tanh\left(\frac{3}{2}z\right)\right], \end{array}$$
where γ∈R is an added time constant that will be implemented in the circuit realization to adjust the difficulty of transitioning between attracting states. For our implementation of Equations (Equation 8)–(10), we let $\gamma =10^{6}$.

## 3. Electronic Physical Realization

Within most applications, smooth continuous-time systems can be realized as electronic circuits comprised of inexpensive components and integrated circuits [30]. In this section, we introduce a comprehensive circuit design to realize Equations (Equation 8)–(10) and construct the system on a breadboard. Experimental recordings of trajectories of interest are then compared with the theoretical system derived in Section 2 as a means to validate the realization. All basic operational amplifiers used are TL082CP, and all individual transistors are MPS2222. In addition, two analog multiplier chips are used, which are the standard AD633 four quadrant multipliers. Note that all schematics shown assume unity gain associated with each multiplier. A complete list of all component values in the following schematics can be found in Appendix A.

### 3.1. Nonlinear Activation Function Circuit

To properly realize Equations (Equation 8)–(10) as an analog circuit, we first must account for the nonlinearity in the system; the hyperbolic tangent function. Previous work has allowed us to easily realize this nonlinearity by means of a simple op-amp and transistor circuit as depicted in Figure 3 [30]. Allowing $V_{in}$ and $V_{o}$ to represent the input and output voltages of the circuit, respectively, Duan and Liao [35] showed that the input-output relation takes the following form:(11)$$V_{0}=-\tanh\left(\frac{R_{2}}{2RV_{T}}V_{in}\right),$$
where $V_{T}\approx 26$ mV is the thermal voltage of the transistors at room temperature. Allowing $R_{2}=520$ Ω, $R_{3}=R_{4}=1$kΩ, $R_{11}=11\;k\Omega$, all other resistors set to R=10kΩ, $V_{CC}=15$ V, and $V_{EE}=-15$ V, reduces the coefficient in front of $V_{in}$ in Equation (Equation 11) to 1, thereby successfully implementing the hyperbolic tangent function as an analog circuit. Further information regarding the error associated with our constructed hyperbolic tangent units can be found in Appendix B.

### 3.2. Schematics of Electronic Birhythmic RNN

Analog circuits can successfully perform addition and subtraction with just operational amplifiers and appropriately connected resistors. The additional use of capacitors allows for an analog implementation of integration [36]. In regards to our realization of Equations (Equation 8)–(10), the analog implementation of the hyperbolic tangent function can be achieved with the schematic demonstrated in Figure 3. In the following schematics, such nonlinear operations are represented by boxes labeled “-tanh”, where terminals IOP1 and IOP2 are the input and output voltage to each hyperbolic tangent unit, respectively. Furthermore, two analog multiplication chips can be used to introduce the phase flow speed dependence on z(t) in Equations (Equation 8) and (9). Thus, the entire system can be constructed entirely from simple analog components.

We begin with the circuit realization of Equation (10), where the schematic is shown in Figure 4. Note that Equation (10) only comprises of the state variable z(t), and can therefore be build independently of Equations (Equation 8) and (9). The variable z(t) is represented as the voltage across capacitor C1 in the provided circuit diagram. The input reading stim_out represents the output of the nonlinear stimulator circuit and will be discussed in Section 4. The schematic of the electronic realization of Equations (Equation 8) and (9) is shown in Figure 5, where the state variables, x(t) and y(t), are represented by the voltages across capacitors C1 and C2, respectively. The “z” input to analog multipliers, M1 and M2, is taken from the integrator output labeled “z” in Figure 4.

### 3.3. Circuit Construction and Experimental Results

A circuit following the schematics depicted in Figure 4 and Figure 5 was constructed on a breadboard. This system was photographed and is depicted in Figure 6. The blue boxes indicate the three separate hyperbolic tangent units. The magenta box indicates the realization of Equation (10), as described by Figure 4, and the green box highlights the two analog multiplier chips and their configuration.

The circuit was constructed so that each limit cycle maintains an approximately constant *z*-component value when sufficiently close to one of the two attractors. Using an four channel oscilloscope, we observed the behavior of the system for different initial conditions as a means to demonstrate stability across runs. These included ten trials initialized randomly within a subset of each of the two basins of attractions. These subsets are (−1.5,1.5)×(−1.5,1.5)×(0,1.5) and (−1.5,1.5)×(−1.5,1.5)×(−1.5,0) for the slow and fast limit cycles, respectively. Each trajectory was given five seconds to relax, and then the state of the system was recorded for five seconds. For the slow limit cycle, the sample mean of the average *z*-component values across all recorded time steps was calculated to be 0.858 V across trials, with a standard deviation of $5.800\times 10^{-3}$ V across those trial averages. For any single trial, the expected standard deviation of the z-component values across all time steps was calculated to be $1.110\times 10^{-16}$ V, suggesting the upper limit cycle is extremely stable. Similarly, for the fast limit cycle, the sample mean of the average z-component values across all recorded time steps was calculated to be −0.907 V across trials, with a standard deviation of $9.248\times 10^{-3}$ V across those trial averages. For any single trial, the expected standard deviation of the z-component values across all time steps was calculated to be $2.133\times 10^{-3}$ V, suggesting the lower limit cycle is also highly stable.

An example of these recordings are depicted in Figure 7 for two different initializations; one within the basin of attraction for each of the two stable limit cycles. Figure 7A,C show the asymptotic behavior of the three state variable voltages in time [x(t),y(t),z(t)]. As intended, the period of the each limit cycles are visibly different, and qualitatively match the asymptotic behavior demonstrated in Figure 2B,C. Furthermore, Figure 7B,D depict the trajectories projected onto the x-y plane, shown in Figure 7A,C, respectively. These recordings indicate the same asymptotic behavior as demonstrated in Figure 1. As such, we can conclude that the analog implementation of Equations (Equation 8)–(10) was successfully realized from the model derived in Section 2.

## 4. Nonlinear Stimulator Circuit Design and Discussion

When looking at the system derived in Section 2, as expressed by Equations (Equation 8)–(10), we notice the geometric simplicity of the global dynamics. As shown in Figure 2A, the two periodic attractors are mirror images of one another across a planar unstable manifold on the x-y plane. This symmetry enabled us to easily decouple the frequencies of each limit cycle by introducing a strictly positive variable time constant dependent only on our *z*-coordinate. However, this introduces a clear problem from the point of view of developing a neurostimulation testbed. In order to transition between basins of attraction, we only need to worry about the component z(t). In other words, transition between attractors can be achieved with constant stimulation on *z*. This solution to attractor transition is trivial, and does not require the use of an intelligent algorithm to solve, rendering it inadequate as a testbed. In order to negate this issue, we develop a state-dependent nonlinear stimulator interface circuit by which all stimulus must pass through. In addition, we allow only one location of stimulation within the circuit previously designed, as marked by the node labeled stim_out in Figure 4, where the output to the stimulator circuit will be fed in and summed with the current value of $\dot{z}$ in the system. By extending the system properly in this way, we can prevent straightforward stimulation patterns (i.e., constant, random, periodic, etc.) from inducing attractor transitions.

Ideally, we want to develop a stimulation circuit such that a pulse stimulated at a random time may cause the output, stim_out, to either increase or decrease. Furthermore, the amount by which the signal can change should exist on a continuum, rather than outputting a voltage from a finite set of values. The final requirement we will enforce for such a circuit will be that a stimulation pulse delivered at a random time should have equal probability of increasing or decreasing the output signal. This last requirement ensures that if one stimulates randomly or continuously, the expected value of stim_out averaged across all time will be zero as time approaches infinity.

Due to the sinusoidal nature of the *x* and *y* components in Equations (Equation 8) and (9), Equation (Equation 12) will serve as the input/output relation of the stimulator circuit at a given time *t*.
(12)$$S_{out}\left(t\right)=S_{in}\left(t\right)x\left(t\right)y\left(t\right),$$
where $S_{out}$ is the output voltage of the stimulator circuit, and $S_{in}$ is the input stimulus (note that $S_{in}=0$ when no stimulus is applied). We note that this system satisfies all of our requirements, as *z* is independent of *x* and *y*. To transition from the slow limit cycle to the fast limit cycle, stimulation must be applied primarily when x(t) and y(t) are of opposite signs. By doing this, the rate of change in z(t) can be made negative on a subset of these intervals (dependent on the current value of z(t)). Assuming that this downward forcing of z(t) can both overcome the pullback from the attractor during these intervals and travel further downward than the accumulated upward travel during periods of no stimulation, then, after a sufficient amount of time, the system will jump over the unstable manifold at z=0 and into the basin of attraction of the fast limit cycle. Since the speed of oscillation is dependent on z(t), any added stimulation will change the time window by which proper stimulation should be applied, ensuring that a periodic stimulation regime will fail to transition between the attractors. By the same argument, to transition from the fast limit cycle to the slow limit cycle, stimulation must be applied primarily when x(t) and y(t) are of the same sign. Figure 8 depicts the schematic for the described stimulator circuit, where $S_{in}$ is labeled as Stimulus.

A physical realization of the stimulator circuit was created in conjunction with the birhythmic system and tested with a stimulus that could be turned on (4 V) or off (0 V). While this voltage range is certainly larger than those seen during neurostimulation, we note that this testbed acts to validate the underlying logic of a stimulation algorithm, and that this range is made proportional, in a practical sense, to the voltage range of the constructed dynamics. For interfacing with a traditional neural stimulation device, an additional signal amplifying circuit will need to be constructed to take in stimulus (typically low current biphasic pulse injections), and control a properly scaled voltage output which will be fed into the input of our designed nonlinear stimulator circuit.

In order to test the validity of our constructed system we recorded twenty trials for each of several stimulation methods, where half of the trials had the state of the system initialized on the slow limit cycle, and the other half of the trials were initialized on the fast limit cycle. Each trial was recorded for five seconds. We show that neither 4 V constant stimulation nor 4 V manually random stimulation induce state transition, and that these results appear invariant to the initial phase angle of the oscillations in x(t) and y(t) when stimulation is applied. Furthermore, we implement a stimulation pattern that can successfully transition between the stable limit cycles.

For constant stimulation, we statistically quantify the expectation of the maximum euclidean distance z(t) moves away from the limit cycle that the system is initialized on for a given trial. The *z* coordinates of both limit cycles empirically derived in Section 3.3 will be used. In the case of the slow limit cycle, the sample mean of the maximum distances achieved on each trial is 0.230 V, with a standard deviation of 0.026 V across trials, implying that each trial was nearly identical in its effectiveness to transition between states. For the fast limit cycle, this sample mean is 0.147 V, and no variance was detectable in these recordings in 24-bit resolution. A 1.5 s segment of z(t) for all trials is depicted in Figure 9, where constant stimulation is denoted in turquoise. Figure 10A,B depict an example of the resultant behavior of the system under constant 4 V stimulation initialized on the slow and fast limit cycles, respectively. The yellow, blue, and pink curves represent the *x*, *y*, and *z* components of the system, and the green curve depicts the voltage over time of stim_out.

For random stimulation, we do the same thing. In the case of the slow limit cycle, the sample mean of the maximum distance achieved on each trial is 0.242 V, with a standard deviation of 0.022 V across trials. For the fast limit cycle, this sample mean is 0.177 V, with standard deviation of 0.013 V across trials. A segment of z(t) for all of these trials is depicted in orange in Figure 9. Figure 11A,B depict an example of the resultant behavior for trajectories initialized on the slow and fast limit cycles, respectively.

Despite executing stimulation at a random initial phase angle in terms of the oscillation in *x* and *y*, the sample statistics suggest a qualitative homogeneity across trials for the two stimulation patterns, which can be seen in Figure 9. In addition, as expected, both of these stimulation regimes do not escape the basin of attraction by which the system is initialized in, as to do so would require the stimulation when the system state is in the proper two quadrants of the x-y plane to overtake the stimulation in the other two for a prolonged period of time. By restricting our stimulation to the desired intervals, we can achieve state transition. In this case, we simply place a 1N4148 diode in the appropriate orientation just prior to the stim_out node in Figure 8 and apply constant stimulation. All of the trials successfully traveled over the separating unstable manifold at z=0 as depicted by the red and blue curves in Figure 9, representing the trials initialized on the slow and fast limit cycles, respectively. An example of the behavior for all system coordinates and the stimulator circuit output is shown in Figure 12 for both trajectories initialized on the slow and fast limit cycles.

Such a demonstration indicates the existence of a stimulation pattern capable of transitioning between basins of attraction in the system. An input stimulus pattern of this form will have to be mimicked by an intelligent algorithm to a sufficient degree of accuracy without the added rectifier, thus aiding in the validation of the use of that algorithm. However, note that a stimulation system would not have direct access to the state variables, the dynamical system, nor how the stimulus modulates the states. Any such algorithm will need to discover the latent dynamics and learn to control the states from observations [32] at the same time it learns to transition out of the current basin of attraction. The continuously changing intervals of when stimulation should be applied, while geometrically simple, are a highly nontrivial relation to uncover within a dynamical system in this manner and should prove challenging for a general algorithm required to learn it.

## 5. Conclusions

For this paper, an electronic testbed for intelligent neurostimulation methods was developed from a physical realization of the dynamical system underlying the architecture of an artificial gated recurrent neural network. Using simple analog components, the system is fabricated such that it exhibits birhythmic behavior, but the stimulation pattern required to transition between attracting states is made nontrivial. As such, standard open-loop stimulation regimes will be unable to induce attractor transitions. We hypothesized that an inability to perform analogous state transitions may underlie some neurological diseases, and prior evidence suggests that, under this hypothesis, global cortical brain function may give rise to very few attractors under the proper spatiotemporal scale of viewing for comatose patients and patients under deep anesthesia. If correct, this enables our system to aid in validating the efficacy of next-generation neurostimulation algorithms upon successfully jumping between basins of attraction. We suspect multiple realizations of this system can be appropriately coupled together to form a more complicated testbed with a wider array of dynamics. Such an extension may be more immediately applicable to more complex neurological functions and dysfunctions. We leave this to future work.

## Figures and Tables

**Figure 1 entropy-22-00537-f001:**
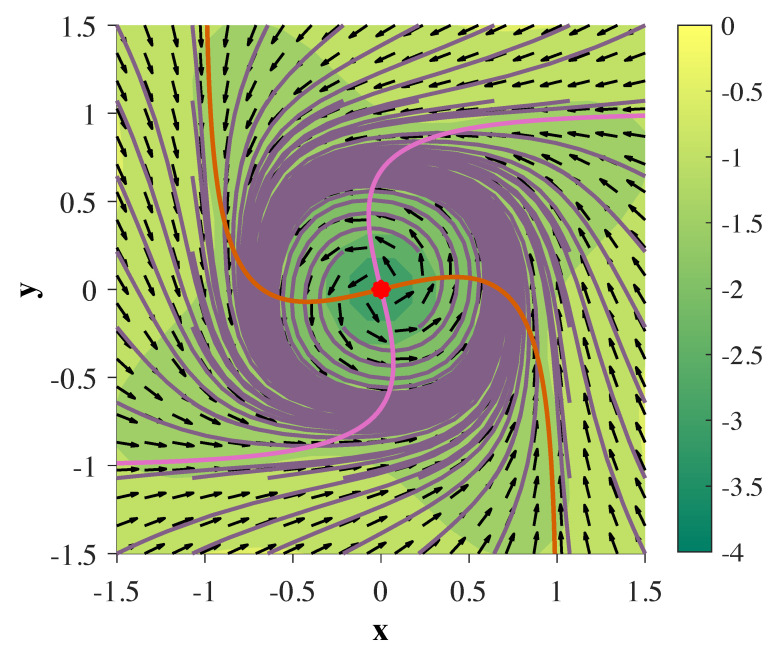
Planar Limit Cycle with 2D continuous-time gated recurrent unit (ct-GRU) depicted in phase space: The red dot indicates an unstable fixed point at the origin unstable, while orange and pink lines represent the x and y nullclines, respectively. Purple lines indicate various trajectories of the hidden state. Direction of the flow is determined by the black arrows, where the colormap underlying the figure depicts the magnitude of the velocity of the flow in log scale.

**Figure 2 entropy-22-00537-f002:**
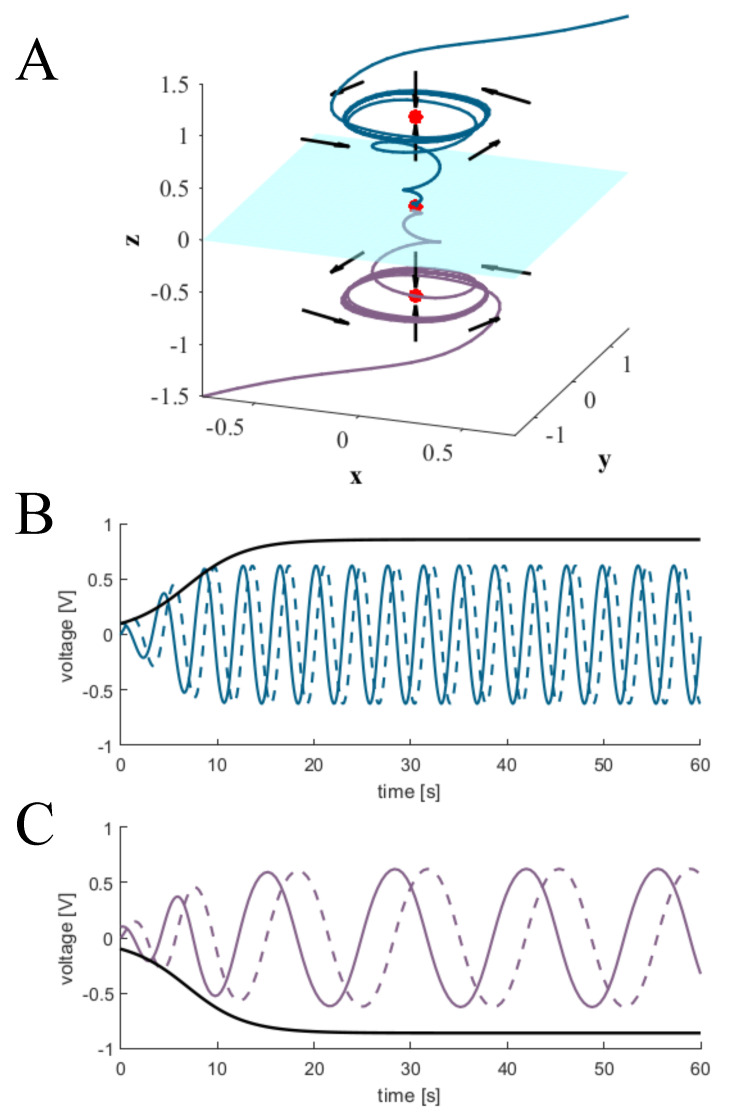
Birhythmicy in 3-dimensions: (**A**): light blue manifold on the x−y plane separates the basins of attraction of the upper and lower limit cycles. Trajectories are colored either dark blue or purple, depending on which basin of attraction they are initialized in. Red dots indicate fixed points, and black arrows depict the direction of flow. (**B**,**C**): *x*, *y*, and *z* components of trajectories initialized in the basins of attractions for the top and bottom limit cycles, respectively. Solid colored lines indicate x(t), dashed lines indicate y(t), and black lines indicate z(t).

**Figure 3 entropy-22-00537-f003:**
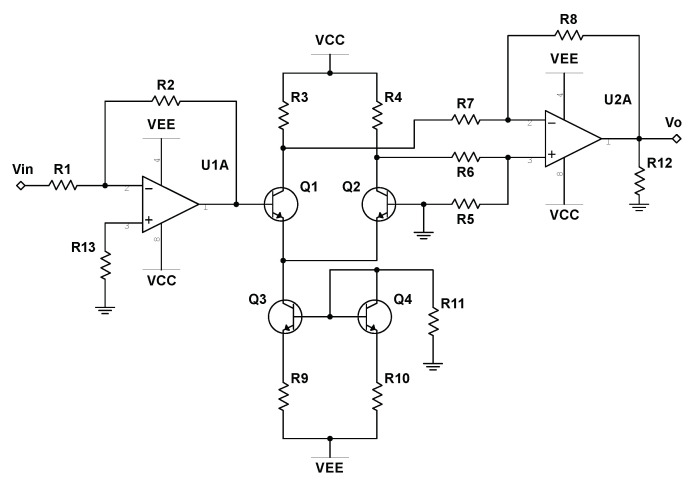
Electronic circuit realization of the hyperbolic tangent function, as implemented in Reference [30]. $V_{in}$ and $V_{o}$ represent the input and output signals, respectively.

**Figure 4 entropy-22-00537-f004:**
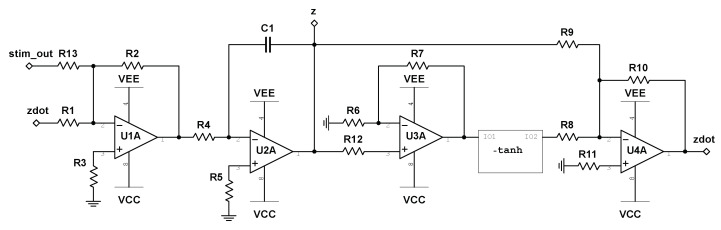
Circuit schematic of $\dot{z}$ for the birhythmic system. The system block labeled -tanh represents the circuit depicted in Figure 3, where I01 and I02 correspond to $V_{in}$ and $V_{o}$, respectively. The terminal labeled stim_out represents the output to the stimulator circuit, as discussed in Section 4.

**Figure 5 entropy-22-00537-f005:**
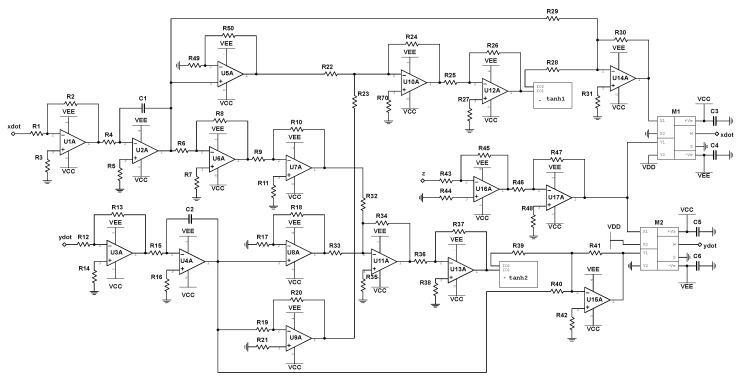
Circuit schematic of $\dot{x}$ and $\dot{y}$ for the birhythmic system. The system blocks labeled -tanh1 and -tanh2 represent the circuit depicted in Figure 3, where I01 and I02 correspond to $V_{in}$ and $V_{o}$, respectively, for both blocks. The two multiplier chips, M1 and M2, are assumed to operate with unity gain.

**Figure 6 entropy-22-00537-f006:**
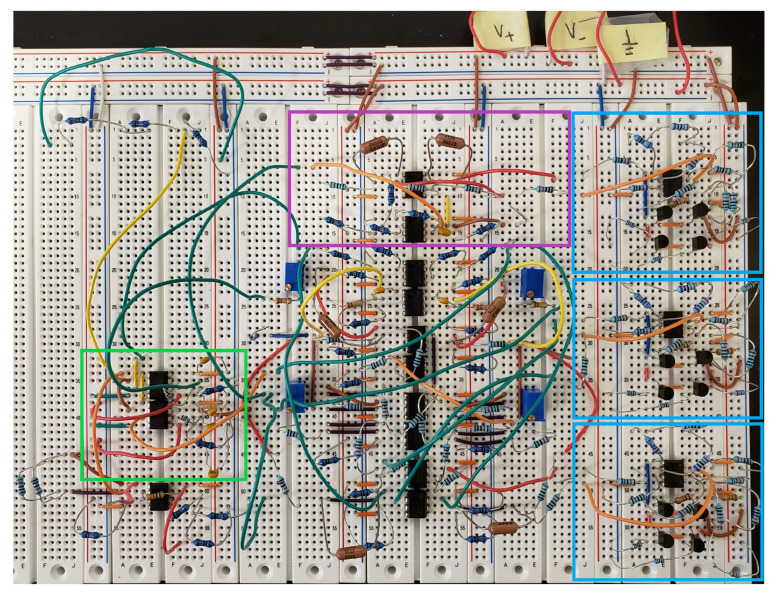
Physical birhythmic circuit constructed on a breadboard. Blue boxes represent hyperbolic tangent units. The magenta box indicates the subsection of the circuit generating $\dot{z}$, and the green box indicates the analog multipliers.

**Figure 7 entropy-22-00537-f007:**
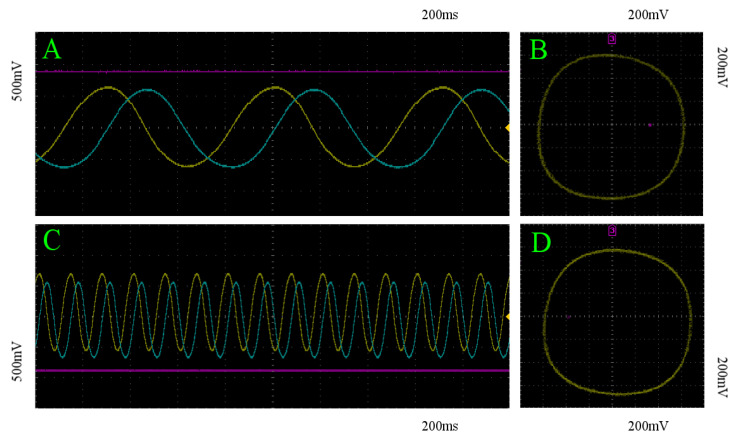
Experimental recordings of the birhythmic circuit: (**A**,**C**): *x* (yellow), *y* (blue), and *z* (pink) with respect to time of trajectories within the basin of attraction of the fast and slow limit cycles, respectively. (**B**,**D**): Projection of the corresponding trajectories in (**A**,**C**) onto the x-y plane, respectively.

**Figure 8 entropy-22-00537-f008:**
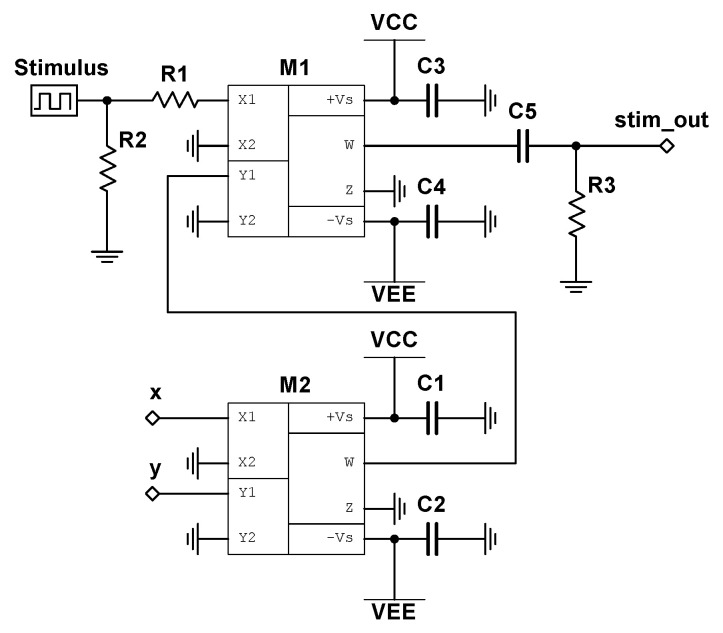
Schematic for nonlinear *stimulator circuit*, with input labeled as *Stimulus*. The output, labeled stim_out, is fed into the terminal with the same name presented in the circuit diagram shown in Figure 4. The two multiplier chips, M1 and M2, are assumed to operate with unity gain, and the *x* and *y* terminals are fed into the equivalently named terminals depicted in Figure 5.

**Figure 9 entropy-22-00537-f009:**
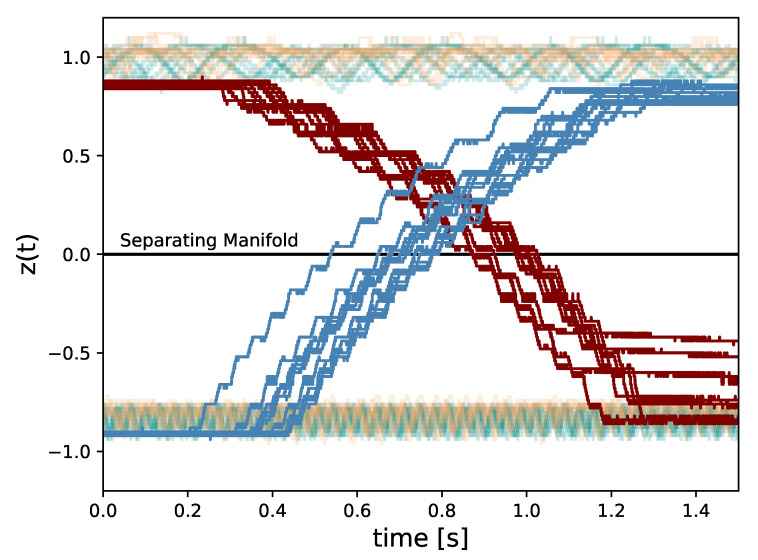
A time window of z(t) for all experimental trials: Red and blue trajectories demonstrate resultant behavior from stimulation patterns designed to transition states from the slow and fast limit cycles, respectively. Turquoise trajectories depict trials of constant stimulation, and orange trajectories show trials of random stimulation.

**Figure 10 entropy-22-00537-f010:**
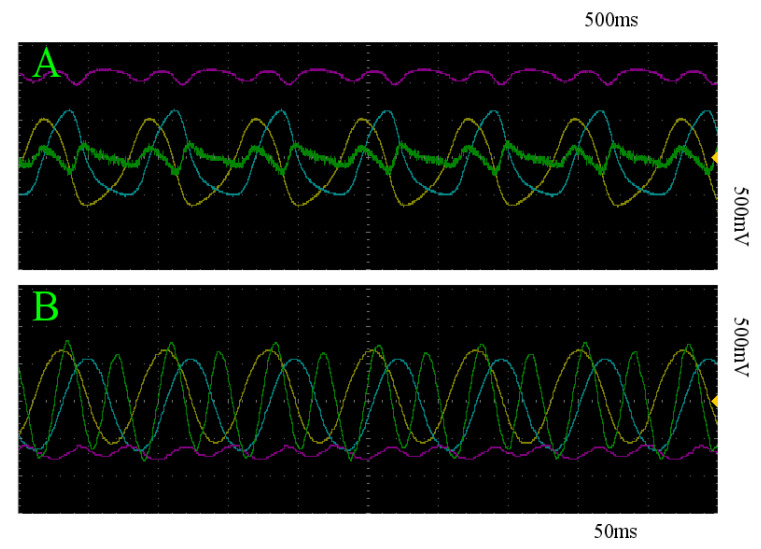
x (yellow), y (blue), z (pink), and *stim_out* (green) with respect to time of trajectories within the basin of attraction of the slow (**A**) and fast (**B**) limit cycles, under constant stimulation. Note that this stimulation regime does not successfully transition between the two attracting states in either direction.

**Figure 11 entropy-22-00537-f011:**
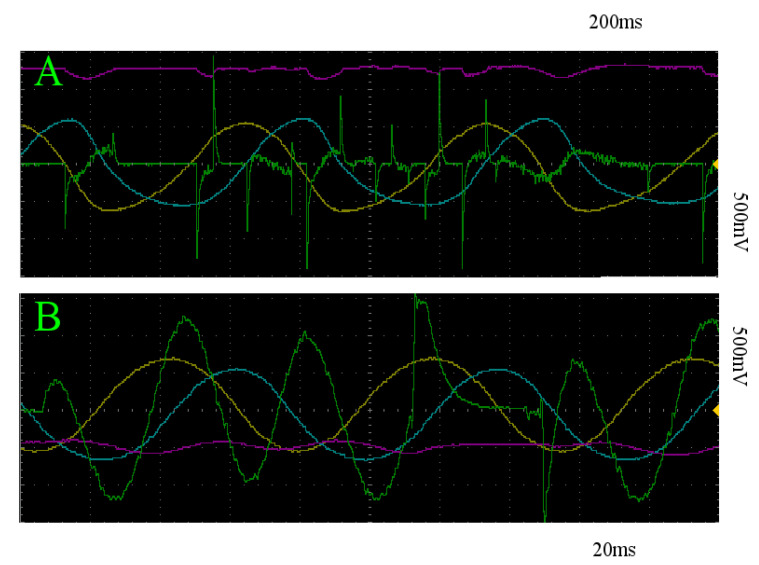
x (yellow), y (blue), z (pink), and *stim_out* (green) with respect to time of trajectories within the basin of attraction of the slow (**A**) and fast (**B**) limit cycles, with random stimulation. Note that this stimulation regime does not successfully transition between the two attracting states in either direction.

**Figure 12 entropy-22-00537-f012:**
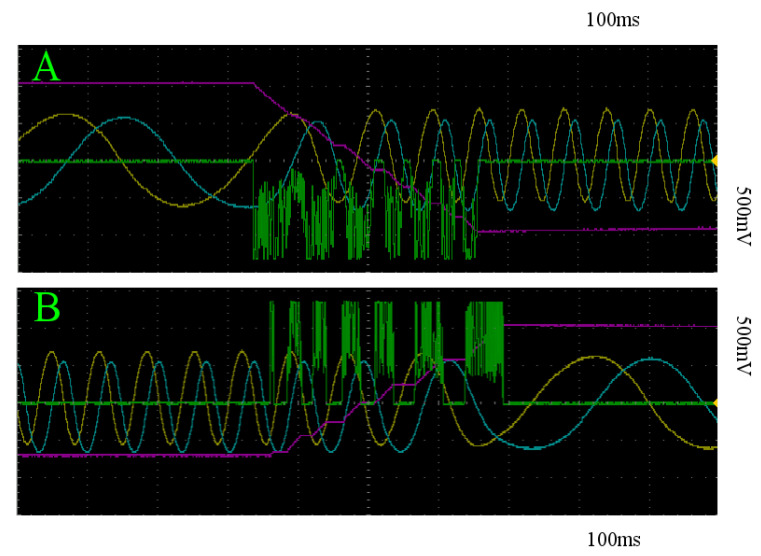
Examples of stimulation patterns capable of inducing transition between states: x (yellow), y (blue), z (pink), and *stim_out* (green) with respect to time of trajectories initialized within the basin of attraction of the slow (**A**) and fast (**B**) limit cycles. As z(t) changes with stimulation so does the frequency of oscillation. As such, the time window to stimulate shifts continuously.

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
