# Peer review of "Birhythmic Analog Circuit Maze: A Nonlinear Neurostimulation Testbed"

_entropy, 2020, doi:10.3390/e22050537_

Round 1
Reviewer 1 Report
The authors designed an analog circuit that can switch between different attractor states. The abstract is written with clinical and neuroscience implications, but the paper itself is disconnected from the clinical and neuroscience literature. I would suggest that the abstract be written to more clearly reflect the work conducted in this manuscript. As written, the clinical impact of the work is unclear.
The paper needs some editing. For examples line 127 is missing a "to" between need and only. Line 137's first my should be may. Lines 146-149 is unclear. Fig 11 caption title is unclear.
Overall, I would suggest creating figures based on data collected from repeated runs of an oscilloscope with some summary statistics versus what appears to be screen captures in Figures 7, 9, & 11 showing replication.
Author Response
Thank you for providing helpful feedback regarding our manuscript. Your comments and suggestions have aided in directing our revisions to improve the readability and desired takeaways of this work.
The following is our point by point response to your review:
Point 1: "The authors designed an analog circuit that can switch between different attractor states. The abstract is written with clinical and neuroscience implications, but the paper itself is disconnected from the clinical and neuroscience literature. I would suggest that the abstract be written to more clearly reflect the work conducted in this manuscript. As written, the clinical impact of the work is unclear."
Response 1: Since the problem we set out to address is new and unexplored, we decided to include the clinical context to the abstract to make it self-contained. We agree that the manuscript is less clear on the clinical impact, though we had made connections to the clinical (deep brain stimulation) and neuroscience literature. Unfortunately, the problem we solve is not a well established one, hence, we allocated more to contextualize the clinical problem that the testbed is designed for. We elaborated on the clinical applications and expanded upon the contexts by which our system is applicable in the introduction and conclusion of the main text of the manuscript. This includes discussion on our system with respect to comatose patients as well as patients under deep anesthesia, where the scale of complexity of the testbed is comparable to the dynamics we hypothesize exist under the correct spatiotemporal scale of viewing.
Point 2: "The paper needs some editing. For examples line 127 is missing a "to" between need and only. Line 137's first my should be may. Lines 146-149 is unclear. Fig 11 caption title is unclear."
Response: Thank you for pointing these out. These grammatical and wording errors have been fixed, the explanation of a stimulation pattern capable of transitioning states has been updated and extended to be more easily understandable, as well as the associated figure caption.
Point 3: "Overall, I would suggest creating figures based on data collected from repeated runs of an oscilloscope with some summary statistics versus what appears to be screen captures in Figures 7, 9, & 11 showing replication."
Response 3: We agree that demonstrating our findings across multiple trials is beneficial and serves to strengthen our claims. We have gone back through the paper and conducted a set of multiple trials for each individual experimental recording originally shown in the submission, along with appropriate statistical measures associated with each, allowing each stimulation trial to apply stimulation at a random initial phase angle of the oscillation represented by x(t) and y(t). We included multiple trials initializing the system at different points within each basin of attraction to demonstrate the stability of the autonomous system, as the trajectories relaxed nearly identically across trials. We used this to validate figure 7 as a consistent example of the autonomous system’s asymptotic behavior. Furthermore, we included an additional figure (now Fig. 9) to section 4. As the successful transition between basins of attraction in our system can be entirely determined by the motion of the z-component of the system in time with stimulation, this figure demonstrates the resultant behavior for z(t) across all trials for each stimulation regime overlayed for a sufficient time window. The qualitative and statistical homogeneity across trials for each stimulation type was demonstrated by showing nearly identical behavior across trials and nearly identical quantified success for each method. This was used to support our use of the previous example figures, which demonstrate an example of each regime for trajectories initialized within each of the basins of attraction.
Reviewer 2 Report
This paper attempts to model the problem of causing brain circuits to undergo state transitions. Here, the authors construct a breadboard realization of a bistable oscillator, and show that it is indeed bistable. They add a set of multipliers that cause stimulation to only be capable of causing transition between basins when the system is in a specific region of each oscillator cycle’s phase space and when stimulation is in a pattern that specifically moves the system along a particular manifold. They show an example where a half-wave rectification can cause state transitions.
The problem is significant; current neurostimulation techniques often fail to place the brain permanently into a healthy state. The demonstrated model is a massive oversimplication of the brain, which would be thousands upon thousands of coupled multistable oscillators. The constructed circuit is a reasonable enough model for a starting point, but far from having practical impact in brain science.
General comments:
- It is still not clear to me why a hardware realization is preferred to software. Given that the system is designed to be well-behaved numerically, it seems fully implementable digitally, which would more easily allow for characterization of a stimulation algorithm across a wide range of parameters.
- It does not appear that this system models real brain stimulation. If I have understood correctly, 4 V constant (DC) stimulation was used. DC currents are indeed applied to brain in some cases, but only at very weak levels. The more common case is square-wave biphasic pulses, which are used for oscillation disorders such as Parkinson disease.
- The exact part number of the rectifier is never given, which would prevent another researcher from replicating the results.
The paper is fine as far as it goes, it adequately describes what was done and what occurred. There should be more said in the discussion/conclusion regarding the points above, especially in terms of this system being limited in how much further it could be taken.
Author Response
Thank you for providing helpful feedback regarding our manuscript. Your comments and suggestions have aided in directing our revisions to improve the readability and desired takeaways of this work.
The following are our responses to each of your points:
Point 1: "The problem is significant; current neurostimulation techniques often fail to place the brain permanently into a healthy state. The demonstrated model is a massive oversimplication of the brain, which would be thousands upon thousands of coupled multistable oscillators. The constructed circuit is a reasonable enough model for a starting point, but far from having practical impact in brain science."
Response 1: Regarding the simplicity of our model, we agree that such a system does not accurately represent the complexity of standard global brain dynamics, at least on a general scale. However, when viewing at different spatiotemporal scales, there can be far fewer attracting states. Take deep anesthesia or comatose patients as an example. Under such conditions, there appears to be coherent synchronization within large brain areas where the model we developed would be more immediately applicable. We add discussion on this in the intro of the main text of the manuscript and updated the conclusion.
Point 2: "It is still not clear to me why a hardware realization is preferred to software. Given that the system is designed to be well-behaved numerically, it seems fully implementable digitally, which would more easily allow for characterization of a stimulation algorithm across a wide range of parameters."
Response 2: The neurostimulators that interface with the neural tissue must have an analog interface where typically current is injected. We built the system in an analog form to ensure that it can be easily interfaced with such stimulation devices. Secondly, an analog representation of our system would lead to inherent imperfections and noise, which in this case is preferable as a means to mimic the unknowability of underlying brain dynamics and forcing the desired algorithms to be more robust. To do so in software would require added complexity to our model, while this complexity we get for free using an analog medium. We added our explanation for this decision in the intro of the main text.
Point 3: "It does not appear that this system models real brain stimulation. If I have understood correctly, 4 V constant (DC) stimulation was used. DC currents are indeed applied to brain in some cases, but only at very weak levels. The more common case is square-wave biphasic pulses, which are used for oscillation disorders such as Parkinson disease."
Response 3: For testing the efficacy of a stimulation algorithm, the circuit described in its entirety within this manuscript is a means to test the logic and implementation of the algorithm to some hardware device capable of admitting a form of electrical stimulation. The stimulus range was made proportional to the scale of the dynamics in realization. For interfacing with actual neurostimulation equipment, the development of an additional segment to the circuit, to be cascaded with the stimulator circuit, would need to be made in order to properly scale (and most likely current control) the input signal. We added this into the discussion in more detail within section 4 of the manuscript.
Point 4: "The exact part number of the rectifier is never given, which would prevent another researcher from replicating the results."
Response 4: Our mistake, the part number is 1N4148 and has been included in the manuscript.
Round 2
Reviewer 1 Report
The manuscript is much improved. One minor comment is that fig 9 appears after fig 10 in the text. Would improve readability to introduce figures sequentially in text.